# Reidentify: Context-Aware Identity Generation for Contextual Multi-Agent Reinforcement Learning

**Zhiwei Xu** [1]  **Kun Hu** [2]  **Xin Xin** [1]  **Weiliang Meng** [3]  **Yiwei Shi** [4]  **Hangyu Mao** [5]  **Bin Zhang** [3]  **Dapeng Li** [3]  **Jiangjin Yin** [6]

## Abstract

Generalizing multi-agent reinforcement learning (MARL) to accommodate variations in problem configurations remains a critical challenge in real-world applications, where even subtle differences in task setups can cause pre-trained policies to fail. To address this, we propose Context-Aware Identity Generation (CAID), a novel framework to enhance MARL performance under the Contextual MARL (CMARL) setting. CAID dynamically generates unique agent identities through the agent identity decoder built on a causal Transformer architecture. These identities provide contextualized representations that align corresponding agents across similar problem variants, facilitating policy reuse and improving sample efficiency. Furthermore, the action regulator in CAID incorporates these agent identities into the action-value space, enabling seamless adaptation to varying contexts. Extensive experiments on CMARL benchmarks demonstrate that CAID significantly outperforms existing approaches by enhancing both sample efficiency and generalization across diverse context variants.

## 1. Introduction

In recent years, the field of multi-agent reinforcement learning (MARL) has advanced rapidly, shifting its focus from theoretical exploration to practical applications. For instance, MARL algorithms have been successfully applied to energy scheduling in power systems (Yan & Xu, 2020), coordinated control in drone formations (Ge et al., 2018),

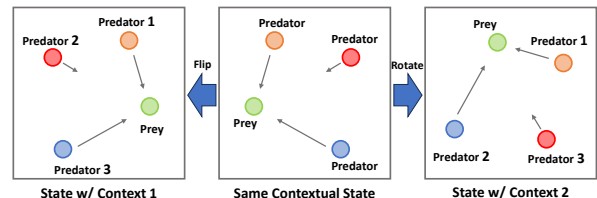

*Figure 1.* An example of CMARL. The central part depicts a fundamental state from the classic predator-prey task. By introducing different contexts to this state, the resulting environmental states on the left and right sides exhibit substantial differences.

and node optimization in communication networks (Mao et al., 2020), yielding promising results. Nevertheless, a significant challenge in reinforcement learning, which is equally prominent in MARL, lies in generalizing across task variants (Kirk et al., 2023). Many existing approaches struggle to maintain comparable performance when faced with even minor task modifications, as the policies trained on original tasks often fail to adapt effectively. Since dynamic problems are common in real-world scenarios, training distinct policies for every variation is infeasible, particularly in multi-agent systems.

Several single-agent studies have focused on generalization across distinct problem variants within the same domain. One approach integrates meta-learning with reinforcement learning to optimize gradient directions that balance multiple objectives (Finn et al., 2017a; Yu et al., 2019; Nagabandi et al., 2019), thereby improving the robustness of reinforcement learning algorithms in multi-task settings. Another line of research leverages transfer learning to fine-tune pre-trained policies (Taylor & Stone, 2009), enabling adaptation to the distributional shifts between original and target tasks. This approach is particularly prominent in robotics, where sim-to-real methods (Zhao et al., 2020) address the discrepancies between simulated environments and the real world. Additionally, inspired by the success of large models (OpenAI, 2023) in natural language processing (NLP), the intelligent decision-making field has begun scaling up policy networks to improve model performance (Reed et al., 2022; Lee et al., 2022). However,

[1]Shandong University [2]National University of Defense Technology [3]Institute of Automation, Chinese Academy of Sciences [4]University of Bristol [5]Kuaishou Technology [6]Huazhong Agricultural University. Correspondence to: Jiangjin Yin <jiangjinyin@mail.hzau.edu.cn>.

*Proceedings of the 42nd International Conference on Machine Learning*, Vancouver, Canada. PMLR 267, 2025. Copyright 2025 by the author(s).

these efforts often face limitations: they either achieve generalization only in simple tasks, narrowly adapt to specific problem variants through fine-tuning, or use model scaling to combine unrelated task strategies. As a result, achieving robust generalization across problem variants remains a significant and unresolved challenge in the field.

Recently, Contextual Reinforcement Learning (CRL) (Benjamins et al., 2023; Hallak et al., 2015) is introduced to formalize the generalization problem across related problem variants. CRL seeks to find a policy capable of solving a group of similar Markov Decision Processes (MDPs), where each MDP presents a variation of the problem, defined by a context variable that reflects the specific characteristics of each variation (Cobbe et al., 2020; Rajan et al., 2023). Similarly, we introduce **Contextual Multi-Agent Reinforcement Learning** (CMARL), which focuses on learning policies for multiple agents across various tasks. These tasks may vary in aspects such as initial states and agent types. For example, while the two initial scenarios may seem distinct, they can be transformed into identical configurations by applying a specific way, as illustrated in Figure 1. Consequently, in CMARL tasks, some related environment states can often be unified into a single contextual state, thereby simplifying the representation of the task. Furthermore, this transformation disrupts the alignment of agent numbering schemes in the two contexts, which underscores the inherent difficulty in ensuring consistent agent identification across different contexts. Current MARL methods cannot automatically assign the appropriate identity to each agent based on the context of the task. As a result, agents must learn separate policies for each situation, significantly reducing sample efficiency in CMARL tasks.

To improve the performance of existing algorithms in the CMARL setting, we propose a novel framework called **C**ontext-**A**ware **ID**entity Generation (**CAID**). The main idea behind CAID is to dynamically assign each agent a unique identity. This is achieved through a context-aware approach integrating global state information with individual agent attributes. By assigning each agent a distinctive identification within the system, the framework effectively reduces the complexity of the contextual decision space. These identities not only serve as unique labels for agents but also provide a foundation for interaction and cooperation among them. Furthermore, the context-aware design ensures that external information about agents is required only during the training phase, allowing for compatibility with most algorithms following the centralized training decentralized execution (CTDE) paradigm. Extensive experiments on benchmarks such as the Vectorized Multi-Agent Simulator (VMAS) (Bettini et al., 2022), Traffic Signal Control (PyTSC) (Bokade & Jin, 2024) and StarCraft Multi-Agent Challenge (SMACv2) (Ellis et al., 2023) demonstrate the effectiveness of CAID.

## 2. Related Work

The generalization of RL algorithms to task variations has garnered significant attention, particularly in the single-agent RL domain. The MAML framework (Finn et al., 2017b), for instance, optimizes initialization parameters to enable models to achieve strong performance on new tasks with minimal gradient updates. First-order optimization algorithms (Nichol et al., 2018), such as Reptile (Nichol & Schulman, 2018), further reduce the computational complexity of MAML, making it more feasible in real-world scenarios. Many meta RL approaches also focus on training history-dependent policies, often implemented as recurrent neural networks (RNNs) (Chung et al., 2014) like $RL^2$ (Duan et al., 2016), which dynamically adapt to interaction histories. MetaGenRL (Kirsch et al., 2020) introduces a meta-learned, low-complexity neural objective function based on diverse agent experiences, leveraging off-policy second-order gradients for improved sample efficiency. Beyond meta RL, transfer RL represents another critical avenue for addressing generalization. For example, a component-based transfer learning framework (Sodhani et al., 2021) leveraging abstract representations has been proposed to facilitate the mastery of complex tasks using prior models. REPAINT (Tao et al., 2021) further enhances deep RL efficiency by combining representation learning and instance transfer techniques. Recent research on in-context learning (Laskin et al., 2023) has explored training models across episodes with task-agnostic processes, enabling generalization to diverse tasks without explicit task-specific adjustments. However, many of these approaches still require fine-tuning to adapt to new tasks, and their applicability is sometimes limited to very simple situations.

Some MARL algorithms enable agents to adaptively define roles or form dynamic groups, allowing them to learn optimal strategies under varying task conditions. REFIL (Iqbal et al., 2021), for example, improves agent robustness to variations in initial states by randomly partitioning environmental entities. ROMA (Wang et al., 2020) and RODE (Wang et al., 2021b) dynamically assign roles to each agent, where each role corresponds to a specific sub-action space, thereby effectively reducing the complexity of the action space. COPA (Liu et al., 2021) employs a global "coach" agent to coordinate partially observable "player" agents through limited information exchange. This design enables COPA to demonstrate zero-shot policy generalization even in teams with dynamic compositions. COLA (Xu et al., 2023) applies the principle of viewpoint invariance from computer vision to map the state space into a discrete representation via contrastive learning. E2GN2 (McClellan et al., 2024) is a framework that improves sample efficiency and generalization through equivariant graph neural networks (GNNs) (Wu et al., 2021). By integrating equivariant and invariant features, E2GN2 overcomes scalability challenges common

in traditional methods. However, these approaches are not specifically tailored for Contextual MARL tasks. In contrast, our proposed CAID assigns an identity to each agent, focusing on maintaining consistent identifiers for corresponding agents across different states. This design minimizes the need for retraining when tasks undergo minor changes.

# 3. Preliminaries

## 3.1. Decentralized Partially Observable Markov Decision Process

In this paper, we investigate a fully cooperative multi-agent task, which can be modeled as a Decentralized Partially Observable Markov Decision Process (Dec-POMDP) (Oliehoek & Amato, 2016). A Dec-POMDP is a framework commonly used to formalize cooperative decision-making problems in partially observable and decentralized environments. Formally, it is defined by the tuple $G = \langle S, U, A, P, r, Z, O, n, \gamma \rangle$, where $S$ represents the set of possible states of the environment, and $A = \{1, \ldots, n\}$ denotes the set of $n$ agents involved. At each time step, each agent $a \in A$ selects an action $u^a \in U$ based on its local observation $z^a \in Z$, obtained through the observation function $O(s, a) : S \times A \to Z$. Here, $s \in S$ denotes the true state of the environment. The joint action of all agents is denoted as $\boldsymbol{u} \in \boldsymbol{U}$. The state transition dynamics are determined by the function $P(s_{t+1}|s_t, \boldsymbol{u}_t) : S \times \boldsymbol{U} \times S \to [0, 1]$. All agents share a common reward function $r(s, \boldsymbol{u}) : S \times \boldsymbol{U} \to \mathbb{R}$, and $\gamma \in [0, 1)$ is the discount factor. The primary objective in the Dec-POMDP framework is to maximize the discounted return $\sum_{j=0}^{\infty} \gamma^j r_{t+j}$.

## 3.2. Value Decomposition Methods

Value decomposition methods (Sunehag et al., 2018; Rashid et al., 2018; Son et al., 2019; Yang et al., 2020) are among the most widely adopted techniques in MARL, particularly for addressing challenges related to coordination. A fundamental concept in value decomposition methods is decomposability, which ensures alignment between global and individual agent objectives. This is formalized through the Individual-Global-Max (IGM) assumption (Son et al., 2019), which stipulates that the optimal action for each agent $\arg\max_{u^a} Q_a^*(\tau^a, u^a)$, must be consistent with the optimal joint action of all agents $\arg\max_{\boldsymbol{u}} Q_{tot}^*(\boldsymbol{\tau}, \boldsymbol{u})$. Mathematically, this condition is expressed as:

$$\arg\max_{\boldsymbol{u}} Q_{tot}^*(\boldsymbol{\tau}, \boldsymbol{u}) = \arg\max_{u^a} Q_a^*(\tau^a, u^a), \quad \forall a \in A, \tag{1}$$

where $\boldsymbol{\tau} \in T^n$ represents the joint action-observation histories of all agents, $Q_{tot}$ is the global action-value function, and $Q_a$ denotes the individual action-value function for agent $a$. Several value decomposition methods have been developed based on this principle, and our proposed CAID

framework can be integrated into these methods to enhance their overall performance.

## 3.3. Contextual Multi-Agent Reinforcement Learning

Contextual Reinforcement Learning (CRL) (Benjamins et al., 2023) extends traditional reinforcement learning by addressing scenarios where a collection of related tasks needs to be solved, with each task influenced by a specific context. The context provides additional information that may not be directly observable but can significantly affect decision-making. CRL formalizes such problems using Contextual Markov Decision Processes (CMDPs) (Hallak et al., 2015), where the state space is augmented to include the environment state and a context variable that captures the conditions under which the agent operates. Formally, the state is represented as $\bar{s} = (s, c)$, where $s$ is the environment state and $c$ denotes the contextual information relevant to the task.

In CMDPs, the context is assumed to remain fixed throughout an episode, ensuring consistency during the agent's interaction with the environment. However, the context can vary across episodes, leading to a distribution of possible contexts $\rho(c)$. Consequently, the initial state distribution in a CMDP is given by $\rho(\bar{s}) = \rho(c)\rho(s|c)$, where $\rho(s|c)$ represents the distribution of the environment state $s$ conditioned on the context $c$. For a fixed context $c$, the CMDP behaves as a traditional Markov Decision Process (MDP). CRL aims to determine an optimal policy $\pi^*$ that maximizes the expected return across all CMDPs. The expected return for a policy $\pi$ within a specific CMDP $\mathcal{M}_c$ is denoted as $\mathcal{R}(\pi, \mathcal{M}_c)$. The overall objective is to maximize this return across the distribution of contexts:

$$\pi^* = \max_{\pi} \left[ \mathbb{E}_{c \sim \rho(c)} \left[ \mathcal{R}(\pi, \mathcal{M}_c) \right] \right].$$

In multi-agent settings, CRL introduces additional complexity, as each CMDP now represents a multi-agent control problem where multiple agents interact both with the environment and with one another. In this paper, we extend CRL to Contextual Multi-Agent Reinforcement Learning (CMARL) by modeling each CMDP as a Dec-POMDP, a framework well-suited for real-world scenarios. The objective of CMARL is to find an optimal joint policy:

$$\boldsymbol{\pi}^* = \max_{\boldsymbol{\pi}} \left[ \mathbb{E}_{c \sim \rho(c)} \left[ \mathcal{R}(\boldsymbol{\pi}, \mathcal{M}_c) \right] \right], \quad \mathcal{M}_c \sim G,$$

where $\mathcal{M}_c$ represents the Dec-POMDP induced by the given context $c$. Our motivation stems from real-world CMARL tasks where the semantics of an environment can shift significantly even within a single episode. For instance, in a traffic control task, the agent behavior required during morning rush hours may differ substantially from that during the evening, despite both being within one episode. We allow the context vector to evolve over time, effectively treating the episode as a sequence of sub-episodes, each governed by a different latent context.

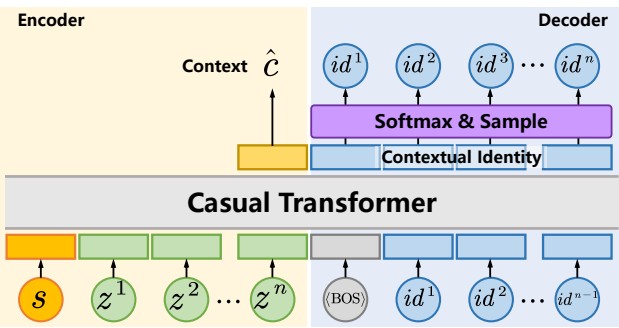

*Figure 2.* The overall architecture of the contextual state encoder and the agent identity decoder. The input includes the global state and observations from all agents, while the output comprises the inferred context and the identity assigned to each agent.

## 4. Methodology

This section presents a detailed overview of the CAID framework, which comprises three core modules. The first two modules are the **Contextual State Encoder** and the **Agent Identity Decoder**, both built upon a Transformer network architecture (Vaswani et al., 2017). These modules work collaboratively to infer contextual information from the environment and generate unique identities for each agent. The third module, the **Action Regulator**, transforms the actions selected by each agent based on its assigned identity, ensuring context-aware decision-making. A key feature of CAID is its ability to map the environment state space into a compact, low-dimensional space. This mapping facilitates the identification of similar contextual scenarios, allowing agents to infer analogous contexts and assign identical identities even when operating in significantly different environment states. By leveraging these identities, agents can effectively reuse previously learned policies, improving adaptability and efficiency in complicated multi-agent environments.

### 4.1. Contextual State Encoder

In practical CMARL tasks, the context $c$ and its distribution $\rho(c)$ are frequently inaccessible, rendering the context-augmented state $\bar{s}$ unavailable during training. As a result, agents must learn their policies based solely on the environment state $s$, which lacks explicit contextual information. When changes in $c$ cause variations in $s$, agents must relearn their policies, which dramatically increases training difficulty.

Inspired by the capabilities of large language models (LLMs) (Touvron et al., 2023), we observe that these models often produce consistent responses even when the order of input words changes. Research indicates that this robustness is attributed to the underlying Transformer architecture, particularly its multi-head attention (MHA) mecha-

nism (Vaswani et al., 2017). This mechanism captures the relationships between input tokens while modeling the data comprehensively from multiple perspectives.

Building on this insight, we propose the **Contextual State Encoder**. It produces a context variable $\hat{c}$ by integrating the environment state $s$ with the local observations $z$ of all agents. This context serves as a substitute for the unavailable ground truth $c$ and concurrently assigns unique identities to each agent.

The structure of the contextual state encoder is illustrated in Figure 2. Inspired by the Seq2Seq architecture (Sutskever et al., 2014), which is widely used in NLP, we employ the encoder component as the contextual state encoder. In CMARL scenarios, the environment state $s$ is highly dynamic, making it challenging to capture contextual information or assign consistent identities based merely on $s$. To address this, we represent the input as a sequence of length $n + 1$, comprising the environment state $s$ and the local observations of all agents $z = \{z^1, z^2, \ldots, z^n\}$. Formally, the encoder $f^{Encoder}$ generates the contextual information as follows:

$$\hat{c} = f^{Encoder}(s, \boldsymbol{z}). \tag{2}$$

While the contextual state encoder does not directly reconstruct the ground truth $c$, it achieves two key objectives: (1) mapping the complex state space into a compact, low-dimensional representation, and (2) extracting relationships between the environment state and agents' local observations. These relationships capture a critical subset of the contextual information, allowing for more efficient policy learning and identity assignment in multi-agent systems.

### 4.2. Agent Identity Decoder

After generating the contextual information, the next step focuses on assigning unique identities to agents. In most existing MARL frameworks (Lowe et al., 2017; Kurach et al., 2020), agent identities are determined using **heuristic rules**. However, in scenarios with similar contextual settings, such methods often fail to ensure consistent alignment of agent identities across different environments. To overcome this limitation, we propose a context-aware identity assignment module capable of automatically assigning each agent $a$ a unique identity $id^a$ based on the generated contextual information.

As shown in Figure 2, the **Agent Identity Decoder** is designed using a causal Transformer decoder. Inspired by natural language generation tasks and the Pointer Network (Vinyals et al., 2015), the identities are determined in an autoregressive manner. The process begins with a special token ⟨BOS⟩, commonly employed in NLP tasks to indicate the start of decoding. Representing the agent identity decoder as $f^{Decoder}$, the probability of identity assignment

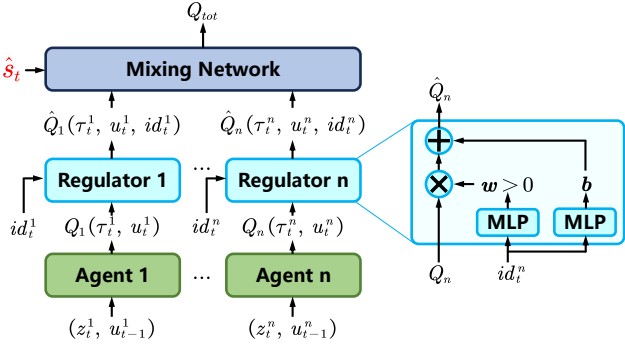

*Figure 3.* Illustration of value decomposition methods with the action regulators.

for each agent $a$ can be expressed as:

$$\mathcal{P}(a) = \text{Softmax}\left(f^{Decoder}(s, \boldsymbol{z}, id^1, \dots, id^{a-1})\right). \quad (3)$$

Each identity $id^a \in \{1, 2, \dots, n\}$ is discrete and must be unique to distinguish agents. Enforcing identity uniqueness is a critical constraint, formally expressed as follows:

$$\forall i \neq j, \quad id^i \neq id^j.$$

To satisfy this constraint, the agent identity decoder employs a dynamic masking mechanism $M$, which ensures that newly generated identities do not duplicate previously assigned ones. The mask $M$ is defined as:

$$M[id] = \begin{cases} 0 & \text{if identity } id \text{ has already been assigned,} \\ 1 & \text{otherwise.} \end{cases}$$

During each decoding step, the probability distribution is dynamically adjusted based on Equation (3) through the application of the specified mask, resulting in:

$$\mathcal{P}(a) = \text{Softmax}(f^{Decoder}(s, \boldsymbol{z}, id^1, \dots, id^{a-1}) \cdot M). \quad (4)$$

To prevent premature convergence to suboptimal identity assignments, identities are sampled rather than greedily selected. Then the identity assignment for each agent $a$ can be formulated as:

$$id^a = \text{Draw}\left(\mathcal{P}(a)\right), \quad \forall a \in \mathcal{A}. \quad (5)$$

This stochastic approach reduces the likelihood of a particular agent consistently dominating a specific identity, promoting more balanced identity assignments among agents.

### 4.3. Action Regulator

After assigning unique identities to agents, the next step is to incorporate this identity information into decision-making processes. In multi-agent systems, agents with the

same identity may have identical absolute semantics in their action spaces, but the relative semantics of their actions can vary significantly across different states. It is necessary to transform their action spaces accordingly.

During the centralized training phase, the contextual state encoder and agent identity decoder generate $\hat{c}$ and $id^a$. However, these identities cannot be used as inputs to the agent networks during the decentralized execution phase. We draw inspiration from hypernetworks (Ha et al., 2017) to overcome this limitation and propose an **Action Regulator** to align agents' action spaces based on their identities. The action regulator operates on the Q-values output by each agent and consists of two linear layers. Both layers take the agent's identity $id^a$ as input, generating the weights $\boldsymbol{w}$ and bias $\boldsymbol{b}$ for a hypernetwork. The transformed Q-value output of the agent network is then expressed as:

$$\hat{Q}_a(\tau_t^a, u_t^a, id_t^a) = \boldsymbol{w}(id_t^a)Q_a(\tau_t^a, u_t^a) + \boldsymbol{b}(id_t^a). \quad (6)$$

By introducing an additional hypernetwork layer to perform an affine transformation on the raw Q-values, the action regulator effectively aligns the action spaces of agents, ensuring that their decisions are consistent with their assigned identities.

To enable seamless integration of the action regulator into existing value decomposition methods for MARL, it must satisfy the Individual-Global-Max (IGM) paradigm, as described in Equation (1). Precisely, the global optimal action should align with the individual optimal actions of the agents. We adopt a simple yet effective approach to meet this requirement: ensuring that $\boldsymbol{w}$ remains positive by taking its absolute value. This guarantees the transformation preserves the consistency between global and individual optimal actions.

In summary, the action regulator serves as a critical component of the CAID framework, enabling identity-based decision-making and ensuring compatibility with existing multi-agent reinforcement learning methods while maintaining theoretical guarantees of the IGM condition.

### 4.4. End-to-End Training

The three main components of CAID, along with their inputs and outputs, have been detailed earlier. The causal Transformer architecture includes a contextual state encoder and an agent identity decoder. The encoder generates a contextual state $\hat{c}$ based on the environment state $s$ and the local observations of all agents $z$. This contextual state $\hat{s} = (s, \hat{c})$ replaces the ground truth $\bar{s}$ and serves as input to the mixing network in value decomposition methods, providing a more comprehensive representation of CMARL tasks compared to the original environment state $s$. The agent identity decoder sequentially predicts each agent's identity in an autoregressive manner. These identities are

subsequently used as inputs to the action regulator, aligning agents' action spaces. The entire CAID framework is trained end-to-end, where the three modules are optimized jointly with the reinforcement learning module. The loss function for this process is as follows:

$$\mathcal{L} = (y_{tot} - Q_{tot}(\boldsymbol{\tau}, \boldsymbol{u}, \hat{s}))^2, \qquad (7)$$

where $y_{tot}$ is the target joint value function, computed as $y_{tot} = r + \gamma \max_{\boldsymbol{u}'} Q_{tot}(\boldsymbol{\tau}', \boldsymbol{u}', \hat{s}')$. All parameters within CAID are optimized by minimizing the temporal-difference (TD) error. A significant challenge arises because each agent's identity is sampled, preventing direct gradient back-propagation to the agent identity decoder. To address this, we utilize **Straight-Through Gradients** (Bengio et al., 2013), a technique easily implemented using automatic differentiation. The agent identity in Equation (5) is modified as follows:

$$id^a = \mathrm{Draw}\left(\mathcal{P}(a)\right) + \mathcal{P}(a) - \mathrm{StopGrad}\left(\mathcal{P}(a)\right), \forall a \in \mathcal{A}, \qquad (8)$$

where the StopGrad function prevents gradient flow through its input during backpropagation. This approach ensures that agent identities are sampled to avoid premature convergence to suboptimal solutions while still allowing gradients to propagate back to optimize the agent identity decoder.

The overall CAID framework is illustrated in Figure 3. By integrating the contextual state $\hat{s}$ and agent identities into value decomposition methods, CAID provides a more expressive representation of CMARL tasks. This design enables CAID to be incorporated into existing value decomposition methods, improving its capability to handle complex multi-agent scenarios.

## 5. Experiments

In this section, we evaluate CAID in three well-known CMARL environments: StarCraft Multi-Agent Challenge (SMACv2) (Ellis et al., 2023), Vectorized Multi-Agent Simulator (VMAS) (Bettini et al., 2022) and Traffic Signal Control (PyTSC) (Bokade & Jin, 2024). The tasks in these domains exhibit considerable variability across episodes, primarily in the agents' positions, agent types, and target locations. First, the performance of CAID is evaluated through a comparison with several classical algorithms, including Weighted QMIX (Rashid et al., 2020), QPLEX (Wang et al., 2021a), and the baseline QMIX (Rashid et al., 2018), along with recently proposed methods such as RIIT (Hu et al., 2021), COLA (Xu et al., 2023), and VMIX (Su et al., 2021). Then the contributions of individual modules within the framework are discussed. To ensure the reliability of the results, each experiment is repeated five times with different random seeds. For a fair comparison, all hyperparameters,

except those introduced specifically by CAID, are kept consistent with the original methods. *Unless explicitly stated, CAID refers to the variant implemented on QMIX.* Details on algorithm hyperparameters are provided in Appendix A.

### 5.1. StarCraft Multi-Agent Challenge

Based on the renowned real-time strategy game StarCraft II, SMAC (Samvelyan et al., 2019) is among the most widely used platforms for multi-agent micro-management experiments. It contains a variety of mini-scenarios, including homogeneous, heterogeneous, symmetric, and asymmetric problem types. Each task requires controlling a team of agents to defeat an AI team controlled by heuristic strategies. Agents can only access information within their local observation range and share a global reward function. However, recent studies (Ellis et al., 2023) have shown that agents in SMAC can achieve near-optimal policies by relying solely on the current timestep and fixed agent IDs, completely disregarding their local observations. This finding highlights a critical limitation of SMAC: a lack of stochasticity, which is essential in real-world problems. To address this limitation, SMACv2 was introduced, where agents' positions and types can change dynamically in each episode, posing new challenges for existing MARL algorithms. We conducted extensive experiments in this classic CMARL environment to evaluate various algorithms, and scenario details are provided in Appendix B.1.

The experimental results in SMACv2 are presented in Figure 4. Across 12 tested scenarios, the inherent stochasticity of the environment occasionally led to scenarios where achieving victory was entirely infeasible, preventing convergence to near-perfect win rates. Our primary observation is that CAID outperforms the baseline QMIX algorithm across all tasks. Furthermore, CAID outperforms other algorithms in most scenarios, particularly in the *Terrain* and *Protoss* series scenarios. In the Zerg series scenarios, the results derived from integrating CAID into QMIX were constrained by QMIX's suboptimal credit assignment capabilities. Nonetheless, CAID consistently demonstrates the fastest initial convergence rate among all tested algorithms.

### 5.2. Vectorized Multi-Agent Simulator

In addition to the StarCraft II game environment, we seek additional effective CMARL environments. The Vectorized Multi-Agent Simulator (VMAS) is a vectorized and fully differentiable simulator designed for efficient MARL benchmarking. It features a high-performance, vectorized 2D physics engine and a diverse suite of challenging multi-robot scenarios. The modular and user-friendly design of the simulator facilitates the creation of custom scenarios, encouraging contributions from the research community. Some scenarios in VMAS represent tasks such as intelligent

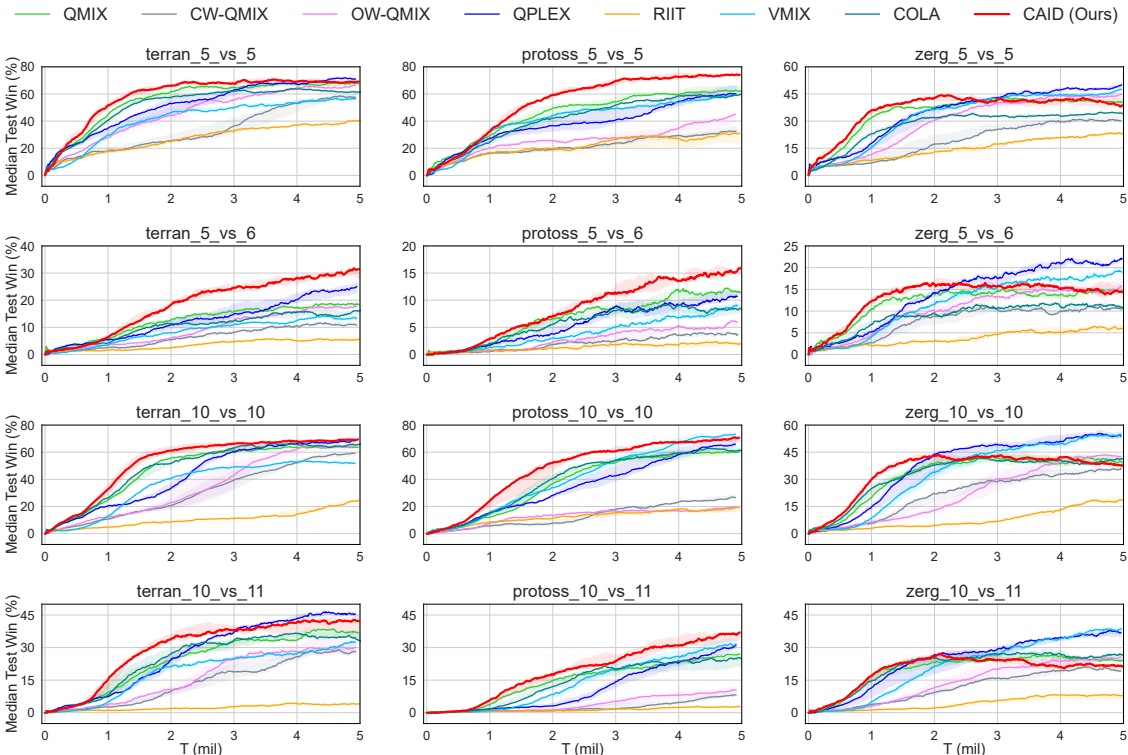

*Figure 4.* Performance comparison with baselines in different SMACv2 scenarios.

formations and obstacle avoidance for drones or uncrewed vehicles, which are of significant practical relevance. Most notably, nearly all tasks can be configured to follow the CMARL setting, where both the initial and goal positions of agents are randomly assigned in each episode. Detailed descriptions of the scenarios employed in VMAS can be found in Appendix B.2.

We used VMAS to compare the performance of three classic value decomposition MARL algorithms, VDN (Sunehag et al., 2018), QMIX, and QPLEX, along with their CAID variants. The learning curves for the CAID variants and their original counterparts are illustrated in Figure 5. It is evident that, in most cases, the CAID variants outperform the original baseline algorithms. This comparison among the three distinct value decomposition methods and their respective variants demonstrates that CAID is applicable to existing value decomposition approaches and can improve their performance. Furthermore, as a distinct value decomposition method, VDN lacks a mixing network module, implying that the contextual state $\hat{s}$ inferred within CAID-VDN is not used. Consequently, the performance improvements observed in CAID-VDN are entirely attributable to the reidentification of agents. This finding underscores the contribution of the agent identity decoder. In the following sections, we systematically evaluate the role and importance of each component in the CAID framework.

## 5.3. Ablation Study

An ablation study was carried out on CAID to evaluate its performance in Traffic Signal Control (TSC) scenarios. PyTSC (Bokade & Jin, 2024) is a development environment tailored for research in traffic signal control, designed to facilitate the rapid development of reinforcement learning-based solutions. In PyTSC, the tasks emulate urban traffic management by controlling traffic lights to optimize vehicle movement through intersections. The project currently integrates with SUMO (López et al., 2018) and CityFlow (Zhang et al., 2019) as simulation backends, offering researchers practical tools to utilize open-source traffic signal control datasets. Notably, it supports the CMARL setting, where the contexts (e.g., traffic flow) in each episode are dynamically randomized. Details regarding the environment are provided in Appendix B.3.

Three variants of CAID were designed, each modifying a single component of the original algorithm. To investigate the importance of the contextual state encoder, we proposed CAID w/o CS, where the contextual state $\hat{s}$ is excluded from the mixing network in the base algorithm. For the agent identity decoder, we introduced CAID w/o AI, in which the agent network does not utilize the newly assigned identity information. Lastly, in CAID w/o ST, new identities for agents are generated using a greedy algorithm instead of

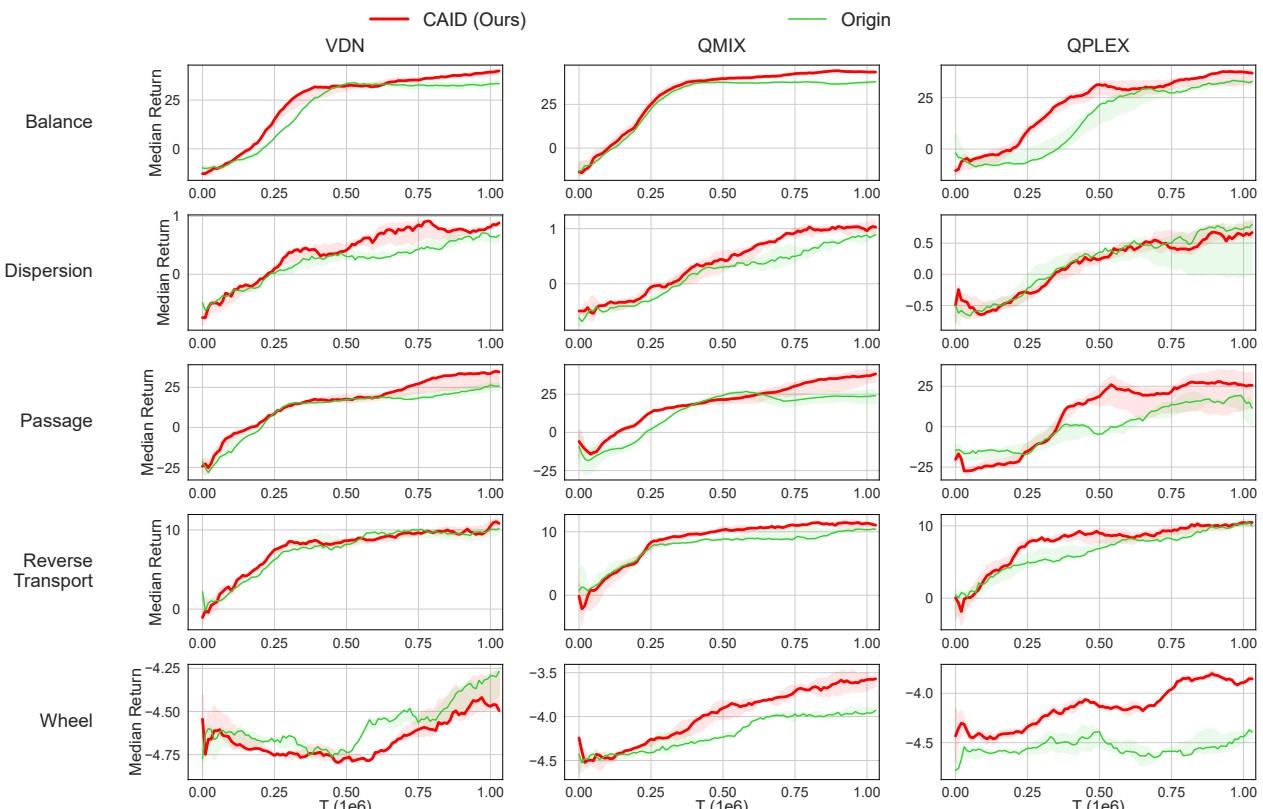

*Figure 5.* Comparison of our approach against baseline algorithms on Vectorized Multi-Agent Simulator.

*Table 1.* Mean of metrics for various algorithms across scenarios in PyTSC.

| Algos | Jinan | | | Hangzhou | | | New York | | |
|---|---|---|---|---|---|---|---|---|---|
| | **Queue** | **Delay** | **Travel Time** | **Queue** | **Delay** | **Travel Time** | **Queue** | **Delay** | **Travel Time** |
| **CAID** | **463.39** | **0.471** | 329.28 | **146.50** | **0.651** | 336.52 | **479.48** | **0.901** | 369.05 |
| **w/o CS** | 476.52 | 0.473 | **326.86** | 155.79 | 0.653 | 340.00 | 488.39 | 0.908 | 372.13 |
| **w/o AI** | 663.59 | 0.498 | 364.05 | 171.37 | 0.659 | 346.77 | 480.41 | 0.902 | **368.38** |
| **w/o ST** | 490.70 | 0.482 | 332.34 | 150.11 | 0.662 | **330.65** | 487.28 | 0.903 | 371.38 |
| **QMIX** | 563.13 | 0.507 | 338.15 | 170.58 | 0.660 | 349.25 | 496.08 | 0.902 | 381.57 |

sampling, potentially leading to suboptimal solutions. The ablation experiment results are illustrated in Table 1. The best performances of the above algorithms are bold.

We selected three key performance metrics that provide a more direct assessment of performance: queue length, delay, and travel time. The results demonstrate that all three ablated variants degrade the performance of CAID. Among them, CAID w/o CS shows the least performance drop, followed by CAID w/o ST. These findings highlight the importance of the identities produced by the agent identity decoder in enabling agents to adapt effectively to dynamic environments. In summary, the designs of the contextual state encoder and agent identity decoder are critical to the effectiveness of CAID.

### 5.4. Visualization

We use a dataset consisting of 20,000 timesteps. The left side of the figure shows the 2D t-SNE embedding of the contextual states produced by the Contextual State Encoder, while the right side presents the 2D t-SNE embedding of the raw states. Each point corresponds to a state, and the color of each point encodes a specific permutation of agent identities in that state. Specifically, we enumerate all permutations of the agent indices $\{0, 1, 2, 3, 4\}$, resulting in $5! = 120$ possible arrangements. Each permutation is mapped to a unique integer in $[0, 119]$ using a bijective function based on Cantor expansion (Lehmer code), enabling us to assign a distinct color to each identity configuration. For comparison, traditional methods typically assign agent identities using heuristic rules, leading to only a single

fixed permutation—represented by a single color in the figure. Notably, the middle of the figure highlights a pair of mirror-symmetric and semantically similar states. In the raw state embedding (right), these states appear distant from each other, whereas in the contextual state embedding (left), they are close together and share similar identity patterns across agents.

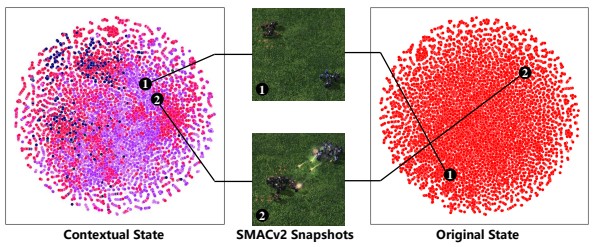

*Figure 6.* The analysis and visualization of the agent identities in the *terrain_5_vs_5* scenario.

## 6. Conclusion

In this paper, we introduce the Context-Aware Identity Generation (CAID) framework, which leverages the global state and local observations from all agents to construct contextual states and dynamically assign agent identities. This design enables agents to adapt effectively to different task variations. The CAID framework can be fully trained in an end-to-end manner using the reinforcement learning paradigm. It is compatible with existing MARL value decomposition algorithms, significantly improving sample efficiency. The extensive experiments in diverse contextual MARL scenarios demonstrate compelling performance, providing strong evidence for the practicality of the proposed framework. We believe this work represents a meaningful step toward bridging the gap between multi-agent reinforcement learning and real-world applications. Promising directions for future research include enhancing the robustness of CAID in contextual MARL tasks with dynamic agent populations and extending its applicability to zero-shot generalization scenarios. Furthermore, identifying a more informative representation of contextual MARL, similar to the agent identities generated by CAID, would be an intriguing avenue for exploration.

## Acknowledgements

This work was supported in part by the National Natural Science Foundation of China under Grant 62302189 and in part by the Fundamental Research Funds for the Central Universities under Grant 2662022XXQD002.

## Impact Statement

This paper presents work whose goal is to advance the field of Machine Learning. There are many potential societal consequences of our work, none which we feel must be specifically highlighted here.

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

## A. Implementation Details

### A.1. Algorithmic Description

The pseudo-code of CAID is shown in Algorithm 1.

---

**Algorithm 1** Context-Aware Identity Generation (CAID)

1: **for** each episode **do**
2:     Get the global state $s_1$ and the local observations $\boldsymbol{z}_1 = \{z_1^1, z_1^2, \ldots, z_1^n\}$ of all agents
3:     **for** $t \leftarrow 1$ to $T-1$ **do**
4:         **for** $a \leftarrow 1$ to $n$ **do**
5:             Select action $u_t^a$ according to the agent network
6:         **end for**
7:         Carry out the joint action $\boldsymbol{u}_t = \{u_t^1, \ldots, u_t^n\}$
8:         Get the global reward $r_{t+1}$, the next local observations $\boldsymbol{z}_{t+1}$, and the next state $s_{t+1}$
9:     **end for**
10:    Store the trajectory in the replay buffer $\mathcal{D}$.
11:    Sample a batch of episodes $\mathcal{B} \sim \text{Uniform}(\mathcal{D})$.
12:    Compute the context variable $\hat{c}$ using Equation (2).
13:    Generate agent identities based on Equation (8).
14:    Evaluate the transformed Q-values for all agents using Equation (6).
15:    Update the parameters of the CAID model using Equation (7).
16:    Periodically update the parameters of the target network.
17: **end for**

---

### A.2. Hyperparameters

Unless specified otherwise, the hyperparameter configurations across different environments are presented in Table 2. These settings are identical to those provided in PyMARL2 (Hu et al., 2021). All experiments in this study were conducted using NVIDIA GeForce RTX 2080 Ti graphics cards and Intel(R) Xeon(R) Silver 4114 CPUs. For all methods, exploration during training is achieved via independent $\epsilon$-greedy action selection, with $\epsilon$ linearly annealed from 1.0 to 0.05 over 50,000 steps. In SMACv2, training ends after 5 million timesteps, whereas it concludes after 1 million timesteps in VMAS and 2 million timesteps in PyTSC.

*Table 2.* Hyperparameter settings.

| Description | Value |
|---|---|
| Learning rate | 0.001 |
| Type of optimizer | Adam |
| How many episodes to update target networks | 200 |
| Reduce global norm of gradients | 10 |
| Batch size | 128 |
| Capacity of replay buffer | 5000 |
| Batch size for parallel execution | 8 |
| Discount factor | 0.99 |

## B. Introduction for Environments

In this paper, we employed three experimental environments: SMACv2, VMAS, and PyTSC. Given the limited research on Contextual MARL and the recent introduction of some new environments, it is essential to provide a detailed overview of these environments tailored to the CMARL setting. The following sections aim to familiarize readers with the objectives of these tasks and the corresponding evaluation metrics, offering a foundation for understanding this emerging field.

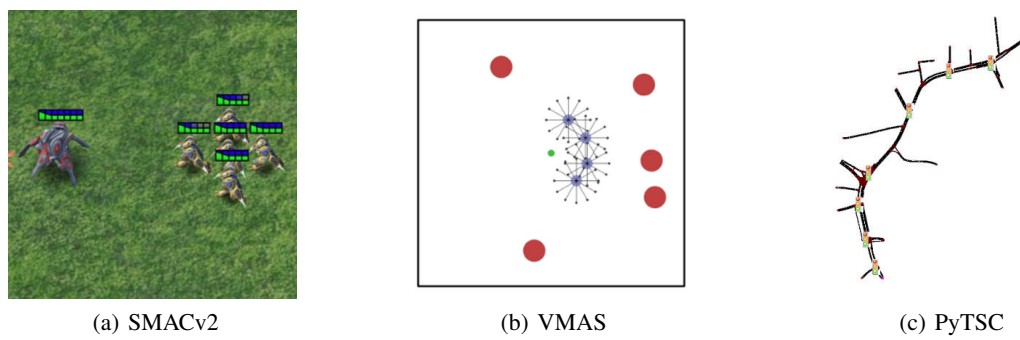

(a) SMACv2           (b) VMAS           (c) PyTSC

*Figure 7.* Screenshots of the two experimental platforms used in this paper.

### B.1. SMACv2

SMACv2 (StarCraft Multi-Agent Challenge version 2) is an enhanced benchmark environment designed to assess the effectiveness of cooperative MARL algorithms. Compared to its predecessor, SMACv2 incorporates procedurally generated scenarios, requiring agents to adapt their collaborative strategies to dynamically changing environments, including diverse enemy configurations and map layouts. This design intensifies the challenges posed by partial observability, compelling agents to base their decisions on real-time sensory input rather than relying on scripted behaviors. In each episode, the positions and types of both ally and enemy agents are subject to change.

The primary focus of SMACv2 is on evaluating the agents' generalization capabilities in previously unseen scenarios. This is achieved through metrics such as win rates, collaboration efficiency, and adaptability in novel environments. By introducing more demanding scenarios and imposing stricter evaluation standards, SMACv2 establishes a rigorous platform for advancing research on multi-agent reinforcement learning algorithms. It fosters algorithmic development in complex, dynamic contexts and ensures robustness and reliability in real-world applications. Descriptions of the scenarios are provided in Table 3.

*Table 3.* Maps in different scenarios.

| Race | Unit Types | Probability of Generation | Scenarios |
|---|---|---|---|
| Terran | Marine
Marauder
Medivac | 0.45
0.45
0.1 | terran_5_vs_5
terran_5_vs_6
terran_10_vs_10
terran_10_vs_11 |
| Protoss | Stalker
Zealot
Colossus | 0.45
0.45
0.1 | protoss_5_vs_5
protoss_5_vs_6
protoss_10_vs_10
protoss_10_vs_11 |
| Zerg | Zergling
Baneling
Hydralisk | 0.45
0.1
0.45 | zerg_5_vs_5
zerg_5_vs_6
zerg_10_vs_10
zerg_10_vs_11 |

### B.2. VMAS

VMAS (Vectorized Multi-Agent Simulator) is an efficient and scalable simulation framework for MARL, featuring a differentiable and vectorized 2D physics engine. VMAS includes a comprehensive suite of complex scenarios that encompass both cooperative and competitive tasks. It supports highly customizable features, including sensors, elastic collisions, rotations, joints, and communication mechanisms. Each scenario integrates tailored reward mechanisms and termination conditions to rigorously evaluate the agents' collaborative strategies and division of labor under diverse task objectives.

VMAS offers a range of pre-built multi-agent reinforcement learning scenarios, addressing both cooperative and adversarial

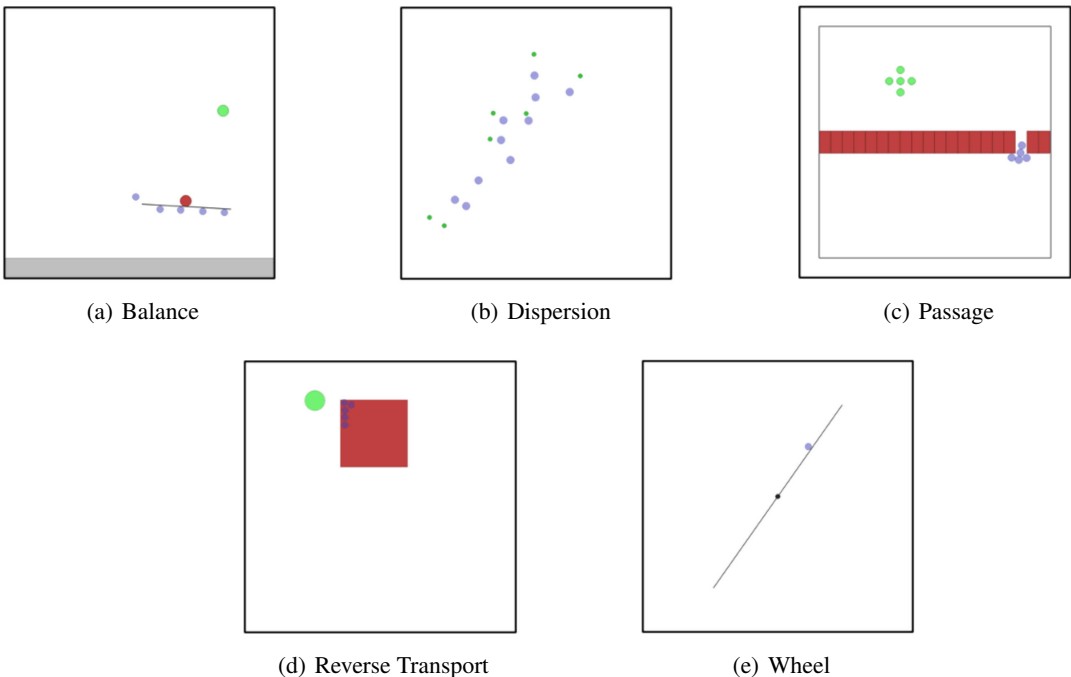

(a) Balance        (b) Dispersion        (c) Passage

(d) Reverse Transport        (e) Wheel

*Figure 8.* Illustration of benchmark tasks in VMAS.

tasks. Descriptions of representative scenarios are as follows:

**Balance.** Agents collaborate to balance a spherical object on a linear platform while transporting it to a designated target. Any fall of the object or platform results in a substantial penalty. Successful completion requires coordinated task allocation to maintain equilibrium while steadily advancing toward the goal.

**Dispersion.** Starting from a common origin, agents must efficiently disperse to distinct target points. This scenario highlights the importance of decentralized coordination, minimizing clustering, and optimizing task completion.

**Passage.** A group of robots must traverse a wall with a narrow passage to reach their assigned destinations. The challenge involves avoiding collisions and orchestrating the use of the passage in a sequential and cooperative manner, testing path-planning capabilities and localized teamwork.

**Reverse Transport.** Agents cooperate to transport a high-mass package to a target location. Given the package's significant weight, individual agents cannot accomplish the task alone. Success demands precise synchronization of force direction and magnitude among all agents.

**Wheel.** Agents encircle a rotating beam anchored at the origin, coordinating their efforts to achieve a target angular velocity. The task requires an optimal distribution of forces to precisely control the beam's speed without overexertion, which could cause deviations from the target.

These scenarios are designed to rigorously evaluate agents' collaborative efficiency, adaptability, and task performance across diverse objectives and constraints. Notably, all scenarios can be configured to the CMARL settings.

### B.3. PyTSC

PyTSC is a simulation environment tailored to multi-agent reinforcement learning (MARL) in the domain of traffic signal control (TSC). Its primary goal is to reduce global congestion across the traffic network, quantified by metrics such as total delay, average waiting time, or vehicle queue length, through the coordinated control of multiple traffic signal agents. In a fully cooperative MARL framework, agents adapt to dynamic traffic patterns and develop efficient control strategies by interacting with both the environment and other agents, ultimately achieving global optimization.

The evaluation of PyTSC focuses on two key dimensions: traffic efficiency and algorithmic performance. Traffic efficiency is

assessed using metrics such as average vehicle waiting time, average travel time, and total network delay, while algorithmic performance is measured by factors like convergence speed and policy stability. Additionally, PyTSC allows for testing the adaptability of algorithms under dynamic traffic flow scenarios, such as morning and evening peak periods, providing a comprehensive assessment of MARL approaches in TSC applications.

*Table 4.* Scenarios in PyTSC environments.

| Scenarios | Simulators | Network Types | Agent Types | Total Agents |
|-----------|-----------|---------------|-------------|--------------|
| Jinan | CityFlow | Real-world | Homogeneous | 12 |
| Hangzhou | CityFlow | Real-world | Homogeneous | 16 |
| New York | CityFlow | Real-world | Homogeneous | 16 |

# C. Additional Results

## C.1. Results of Dynamic Role Assignment Methods

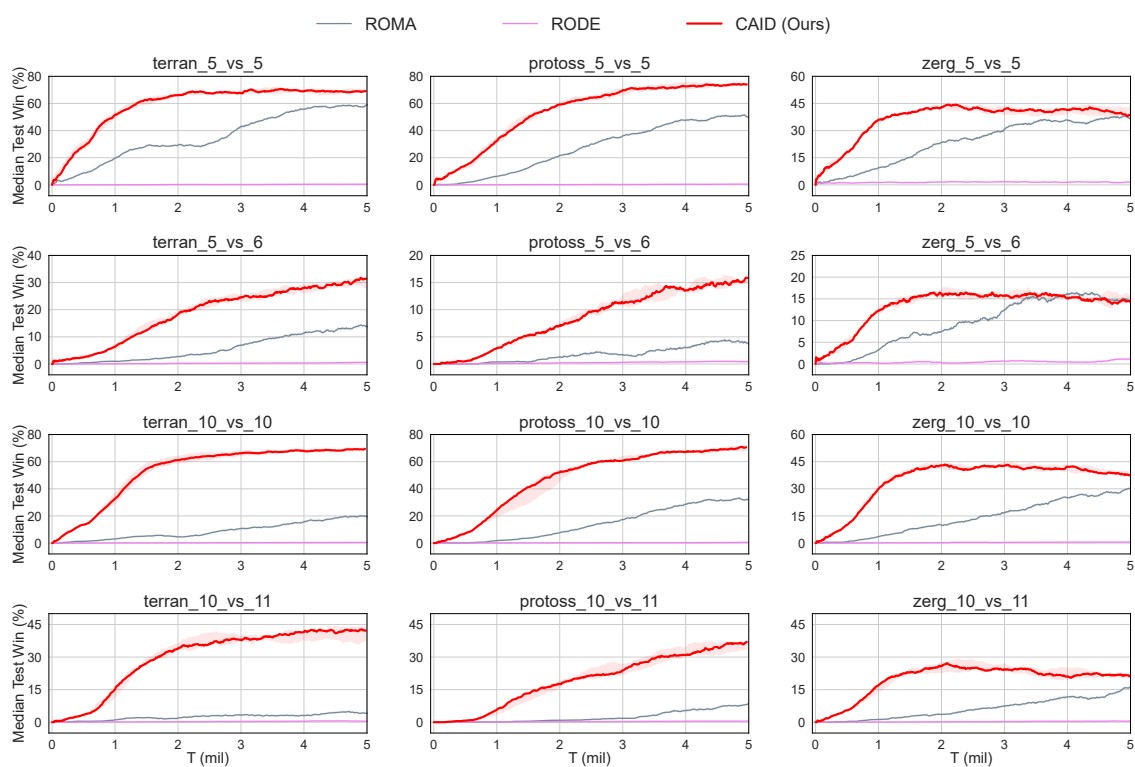

*Figure 9.* Performance comparison with dynamic role assignment methods in different SMACv2 scenarios. We selected ROMA and RODE as two representative methods for role assignment. All methods were configured with identical hyperparameters, including the optimizer and techniques such as eligibility traces, to ensure consistency with other algorithms implemented in PyMARL2. It is worth noting that RODE fails to perform properly in the SMACv2 environment. This limitation arises because the original RODE implementation modifies the SMAC environment by sorting enemy units in the environment file. Such changes simplify the original SMAC environment, which undermines its complexity and makes the implementation incompatible with the more challenging and standardized SMACv2. As a result, RODE is not applicable to the Contextual MARL setting. CAID achieves superior performance and faster convergence.

## C.2. Additional Ablation Study

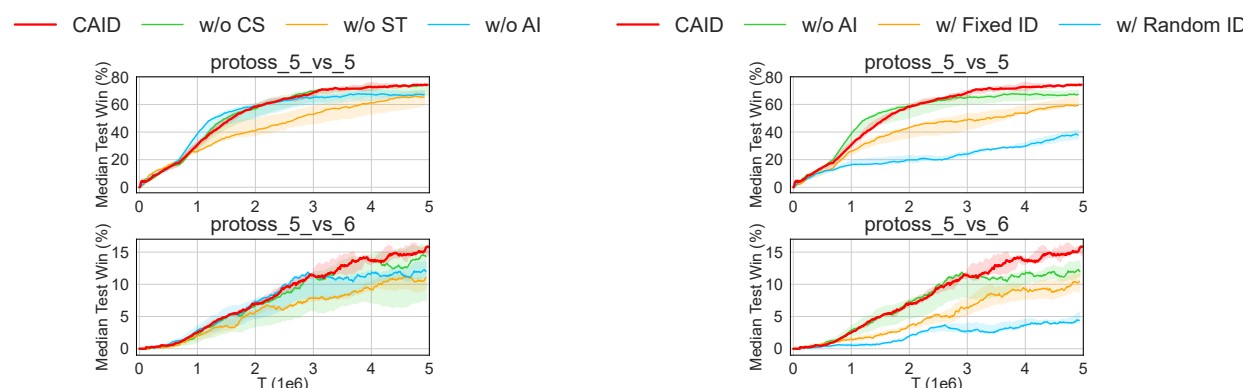

*Figure 10.* **Left:** The learning curves of the three variants discussed in Section 5.3 on the SMACv2 benchmark. CAID w/o AI refers to the variant where only the Action Regulator component of CAID is removed, while all other components remain unchanged. The results demonstrate that all three variants underperform compared to the full CAID model, highlighting the essential role each module plays in the overall performance of CAID. **Right:** The learning curves of different variants for identity modeling, including ablation and alternative strategies. CAID with Fixed ID denotes a configuration where each agent is assigned a static identity based on its index in the agent list. CAID with Random ID refers to a setting where agents are assigned randomly generated but episode-consistent identities.

