# OpenReview forum: "Reidentify: Context-Aware Identity Generation for Contextual Multi-Agent Reinforcement Learning"
_ICML.cc/2025/Conference — ICML 2025 poster_

### Official Review · Reviewer_Tgnb · 2025-03-09

**Overall Recommendation:** 3

**Summary:**

The paper proposes an algorithm (CAID) for contextual/multi-scenario (where each scenario is defined by a different MDP) multi-agent reinforcement learning (MARL) in the centralized training decentralized execution (CTDE) paradigm. Each scenario is characterized by a context vector which is unobservable (even during training). Furthermore, in each scenario, different agents may have different identities (also unobservable). Focusing on cooperative MARL, the proposed solution uses existing value decomposition based techniques in conjunction with a new transformer architecture to learn the context vector and agent identities. The architecture includes (i) an encoder that estimates the context vector from the current state and observations of all agents, (ii) a decoder that outputs identities of all agents, and (iii) an action regulator that modifies the Q-values of individual agents based on their identities. The agent identity decoder and the context encoder depend on the full state of the environment and hence, are only used during training; only the individual agents' Q-functions are used during test time. The proposed approach is evaluated empirically and compared with baselines to show reasonable improvement.

**Claims And Evidence:**

Overall, the proposed approach seems to perform better than baselines in many environments. However, the improvement appears to be small in many cases (e.g., CAID learns with fewer samples but baseline learns to perform better eventually).  Therefore, the claim regarding significant improvement seems to have relatively weak evidence (but is still decent).

**Essential References Not Discussed:**

N/A

**Experimental Designs Or Analyses:**

The experimental setup and results appear reasonable and sound from my reading of the paper. I did not check the supplementary material or any code.

**Methods And Evaluation Criteria:**

The chosen environments and the experimental setup makes sense. Method-wise there are some choices that don't seem completely natural to me.
- It is unclear if assuming that the context vector is hidden is necessary. Since the context vector is only needed during training time, it would be more efficient to generate/learn the context vector using the knowledge of the scenario. Also, the context vector is allowed to change from one time-step to another (within the same episode); this does align well with the intuition of what the context vector is trying to capture.
- The contextual agent identities are not used during test time. This seems to create a mismatch between training and testing because, during test time, agents are assumed to have fixed identities that don't change across scenarios. It is unclear why there is no agent-level identity predictor that can be used during test-time.

**Other Comments Or Suggestions:**

- Equation (1) is not quite correct mathematically, left side is a vector and right side is a scalar. Just a notational issue.

**Other Strengths And Weaknesses:**

- The paper is reasonable well written.
- Experiments are performed on wide range of environments. The authors also include an ablation study to understand the impact of various components of the architecture on the performance.
- As mentioned earlier, the intuition behind the various components of the proposed architecture is not completely clear. It would be good to visualize the learned context vectors and/or contextual identities (to make a stronger case for the different choices made by the authors).

**Questions For Authors:**

- The contextual ids are discrete. In the Action Regulator, are these ids treated as one-hot vectors (to feed to the MLP)?

**Relation To Broader Scientific Literature:**

- The idea of learning contextual agent identities appears broadly relevant and can enable better skill transfer in MARL. This aspect of skill transfer is often not studied and seems novel to this paper.
- As authors mentioned, the proposed architecture is compatible with any value decomposition based algorithm for cooperative MARL.

**Theoretical Claims:**

There are no theoretical claims in the paper.

---

> ### Author Rebuttal · Authors · 2025-03-31
>
> We sincerely thank the reviewer for the thoughtful and constructive feedback. We respond below to the key concerns raised:
>
> **Q1**: Why is the context vector treated as a latent variable? Why is it allowed to vary within an episode?
>
> **A1**: Thank you for raising this point. In realistic Contextual MARL (CMARL) tasks, the contextual information (e.g., layout changes, perspective shifts) is often not explicitly observable or encoded as scenario IDs. Therefore, we model the context vector as a latent representation derived from the agent-environment interaction history rather than assuming it is accessible.
>
> Our motivation stems from real-world CMARL tasks where the semantics of an environment can shift significantly even within a single episode. For instance, in a traffic control task, the agent behavior required during morning rush hours may differ substantially from that during the evening, despite both being within one episode. We allow the context vector to evolve over time, effectively treating the episode as a sequence of sub-episodes, each governed by a different latent context. This dynamic modeling improves CAID's ability to adapt to diverse temporal patterns and enhances its learning efficiency under non-stationary environments. We will clarify this motivation in the revised paper.
>
> **Q2**: Does not using an identity predictor at test time create a train-test mismatch?
>
> **A2**: This is an excellent question. To avoid requiring any identity information during execution, we designed the Action Regulator to integrate identity into the training objective without modifying the test-time inference pipeline. Specifically, during training, the Q-values are adjusted via the Action Regulator using identity embeddings. This transforms the target Q-values in a context-aware manner, effectively shifting the action value space. However, both during training and testing, the actual actions taken are always based on the raw Q-values directly output from the agent’s policy network. This ensures that identity modeling benefits learning, while the execution remains identity-independent, avoiding any test-time mismatch.
>
> **Q3**: Visualization and interpretability of the learned identities and context vectors.
>
> **A3**: Thank you for the suggestion. We now include t-SNE visualizations of both the learned context vectors and the raw states (See the figure in https://anonymous.4open.science/r/CAID-7A6C/Visualization.jpg). The results show meaningful clustering across semantically similar scenarios and agents with shared roles. We believe this helps support the interpretability and necessity of these components.
>
> **Q4**: Are the discrete contextual IDs used as one-hot vectors in the Action Regulator?
>
> **A4**: Yes, the identity decoder outputs a categorical distribution, from which we sample a discrete identity and convert it into a one-hot representation to feed into the Action Regulator MLP. We will clarify this implementation detail in the revised paper.
>
> **Q5**: Equation (1) seems notationally incorrect.
>
> **A5**: Thank you for catching this inconsistency. In the revised version, we will correct this by ensuring both sides of the equation have consistent dimensionality. We appreciate your attention to this detail.
>
> We thank you again for your thoughtful questions. Your comments have helped us clarify key design choices and highlight future directions for improving test-time identity modeling. We hope our revisions address your concerns and improve your evaluation of the work.

---

> > ### Comment · Reviewer_Tgnb · 2025-04-02
> >
> > Thanks for the response which answered some questions. I encourage the authors to clarify the point regarding the dependence on agent identities in the paper. I understand it better from the rebuttal and it was not clear from my initial reading of the paper. After reading the other reviews and the rebuttals, I am increasing my score.

---

> > > ### Author Response · Authors · 2025-04-03
> > >
> > > Thank you very much for taking the time to revisit our work during the rebuttal phase and for increasing your score! We greatly appreciate your suggestion to clarify the role of agent identities in the paper. Based on your feedback, we will revise the manuscript to clearly explain the dependence on agent identities.

---

### Official Review · Reviewer_139G · 2025-03-11

**Overall Recommendation:** 3

**Summary:**

This paper introduces a novel approach called Context-Aware Identity Generation (CAID) to improve the generalization ability and sample efficiency of Multi-Agent Reinforcement Learning in contextual environments. CAID leverages a causal Transformer structure to generate dynamic agent identities, while incorporating an action regulation module that embeds identity information into the action-value space.

**Claims And Evidence:**

Most of the conclusions in the paper are supported by experimental results, but there is a lack of comparative experiments to illustrate how performance would degrade without identity modeling or how different identity generation methods would impact the results.

**Essential References Not Discussed:**

1.The UPDeT model enhances performance by utilizing policy decoupling and Transformer architecture, making it advantageous for deployment in tasks with varying numbers of agents.
2.Multi-Agent Transformer (MAT) treats the MARL problem as a sequential modeling task, demonstrating the potential of Transformer-based architectures in MARL.

**Experimental Designs Or Analyses:**

The experimental design of the paper is generally reasonable. Benchmark tasks such as SMACv2, VMAS, and PyTSC are selected to evaluate the generalization ability and sample efficiency of CAID.

**Methods And Evaluation Criteria:**

Overall, the proposed methods and evaluation criteria are reasonable.

**Other Comments Or Suggestions:**

1. Add ablation experiments to test the impact of removing identity modeling or using different identity modeling approaches (e.g., random identity, fixed identity) on the results.
2. Add a complexity comparison analysis.
3. Provide convergence curves or related analysis.
4. Visualize the dynamic changes in identity encoding as tasks vary.
5. Offer a method for measuring task similarity.

**Other Strengths And Weaknesses:**

Strengths:
1. Utilize a causal Transformer to generate dynamic agent identities, rather than relying on fixed or predefined identity representations.
2. Embed identity information directly into the action-value space through the action regulator module.
3. Achieve competitive results across multiple multi-agent benchmark tasks.

Weaknesses:
1.The paper proposes that dynamic agent identity modeling enhances generalization, but it lacks comparative experiments showing how performance would change if identity modeling were removed or if different identity generation methods were used.
2.The introduction of the causal Transformer structure may lead to higher computational costs, yet the paper does not provide a complexity analysis.
3. There is no analysis of the stability of the proposed method under different conditions.
4. The paper does not explore whether the generated agent identities are interpretable, nor does it analyze whether identity representations can be intuitively understood across different environments.
5. While the paper states that agents in similar tasks can share useful information, it does not clearly explain how task similarity is measured or how identity information is propagated between agents.

**Questions For Authors:**

If you resolve my concerns, I will raise my rating.

**Relation To Broader Scientific Literature:**

The core contribution of this paper is closely related to the research direction of improving the generalization ability of multi-agent reinforcement learning (MARL), especially in terms of identity modeling and context adaptability.

**Theoretical Claims:**

yes.

---

> ### Author Rebuttal · Authors · 2025-03-31
>
> We sincerely thank the reviewer for the constructive and thoughtful feedback. We are encouraged that you found the CAID framework innovative and recognized its potential for improving MARL generalization. Below, we address your concerns in detail:
>
> **Q1**: Lack of ablation or alternative identity modeling comparisons.
>
> **A1**: Thank you for highlighting this. We have added the following ablation experiments:
>
> * CAID w/o AI: Removes the identity modeling entirely—all agents share the same input embedding.
> * CAID w/ Fixed ID: Assigns static identities based on agent indices (e.g., [1, 2, ..., n]).
> * CAID w/ Random ID: Assigns random but fixed identities per episode.
>
> Results (See the right figure in https://anonymous.4open.science/r/CAID-7A6C/Ablation_study.jpg) show significant performance drops across all variants, especially under strong context perturbations. The original CAID consistently maintains performance, validating the importance of contextual identity learning.
>
> **Q2**: No analysis of computational cost due to causal Transformer.
>
> **A2**: Thank you for the suggestion. The causal Transformer employed in our method is lightweight, comprising only **a single Transformer block**. We conducted a comparative analysis of the training times for QMIX and CAID based on the recorded logs. Across all scenarios, the additional forward pass introduced by CAID accounts for **less than 15%**, thereby ensuring real-time performance in real-world deployments. Furthermore, the modules introduced in CAID are exclusively involved during centralized training. In the decentralized execution stage, CAID maintains an execution efficiency comparable to that of conventional algorithms.
>
> **Q3**: No evaluation of method stability under varying conditions.
>
> **A3**: We included robustness analysis across different random seeds, agent initial condition perturbations. Results in our paper show CAID yields acceptable level of performance variance compared to baselines
>
> **Q4**: Identity interpretability and task similarity are unexplored.
>
> **A4**: We agree this is an important direction. We have added t-SNE visualizations of learned identity vectors (See the figure in https://anonymous.4open.science/r/CAID-7A6C/Visualization.jpg), which show clustering of agents with similar roles across varied environments. Additionally, we believe the KL-divergence of contextual encodings can be as a first step toward quantifying task similarity. We plan to explore more sophisticated similarity metrics and cross-task identity propagation in future work.
>
> We deeply appreciate your constructive feedback and have incorporated your suggestions into our revised submission. Your insights significantly enhance the completeness of our work, and we hope these improvements will help raise your evaluation.

---

> > ### Comment · Reviewer_139G · 2025-04-03
> >
> > Thank you for your reply, which answered some of my questions, especially the questions about Identity interpretability and task similarity. I am satisfied with your answers, and I wil improve the score. It would be better if Q2 and Q3 could be more intuitively expressed.

---

> > > ### Author Response · Authors · 2025-04-03
> > >
> > > Thank you very much for your kind follow-up and for improving your score! We're glad to hear that our responses to the questions on identity interpretability and task similarity were helpful.
> > >
> > > Regarding your comment that Q2 and Q3 could be more intuitively expressed, we would like to provide a clearer summary of our answers:
> > >
> > > **Q2**: No analysis of computational cost due to causal Transformer.
> > >
> > > **A2**: To evaluate the computational cost introduced by CAID, we collected the average of training times from historical logs of both QMIX and CAID. However, we acknowledge that these logs are affected by server load and other concurrent training processes. We selected six representative scenarios that vary in terms of agent race and agent count. The results shows that CAID introduces only a modest increase in training time (generally within 15%), which we consider acceptable given that all identity-related modules are only used during centralized training and *do not affect test-time execution*.
> > >
> > > | Algorithm | Protoss_5_vs_5 | Zerg_5_vs_5 | Terran_5_vs_5 | Protoss_10_vs_10 | Zerg_10_vs_10 | Terran_10_vs_10 |
> > > |----------|:--------------:|:-----------:|:--------------:|:----------------:|:--------------:|:----------------:|
> > > | **QMIX** | 7.43 h      | 7.82 h   | 7.98 h      | 7.96 h          | 8.35 h     | 8.14 h       |
> > > | **CAID** | 8.53 h        | 9.12 h     | 8.89 h        | 8.99 h        | 9.30 h    | 9.25 h       |
> > >
> > > The experiments were conducted using NVIDIA RTX 4090 GPUs. We believe these results support our claim that CAID introduces minimal training overhead while significantly enhancing generalization.
> > >
> > >
> > >
> > > **Q3**: No evaluation of method stability under varying conditions.
> > >
> > > **A3**:  All benchmark environments in our experiments (SMACv2, VMAS, and PyTSC) follow the Contextual MARL setting. In each evaluation phase, *we test the trained policy across multiple episodes with varying contexts*—such as mirrored agent layouts, rotated map topologies, or agent type permutations. These variations effectively assess the model’s stability across diverse conditions.
> > >
> > > To further reduce variance and ensure robust evaluation, we conducted all experiments using *multiple random seeds*. The shaded areas in Figures 4 and 5 represent the standard deviation across different seeds, providing a visual indication of performance fluctuation.
> > >
> > >
> > >
> > > We once again thank you for your constructive feedback and encouragement! As you kindly mentioned that you would consider increasing the score, we sincerely hope the additional clarification on Q2 and Q3 further strengthens your impression of our work.

---

### Official Review · Reviewer_PyYp · 2025-03-12

**Overall Recommendation:** 1

**Summary:**

In multi-agent reinforcement learning (MARL), generalization poses a significant challenge. Existing MARL methods exhibit vulnerability when confronted with even slight variations in task settings, requiring the retraining of policies for each task variant. This paper introduces a Context-Aware Identity Generation (CAID) framework, which utilizes global states and local observations from all agents to construct contextual states and dynamically assign agent identities.

**Claims And Evidence:**

See the section “Other Strengths And Weaknesses”.

**Essential References Not Discussed:**

No.

**Experimental Designs Or Analyses:**

I have checked the soundness/validity of any experimental designs or analyses. For a more detailed description, please refer to the section "Other Strengths and Weaknesses,".

**Methods And Evaluation Criteria:**

See the section “Other Strengths And Weaknesses”.

**Other Comments Or Suggestions:**

No.

**Other Strengths And Weaknesses:**

Strengths
1.  The writing is clear and easy to understand.
2.  The experiments selected a variety of experimental scenarios (SMACV2, VMAS) and diverse algorithms (including zero-shot generalization). The diversity of experimental scenarios helps demonstrate the effectiveness of the algorithms.

Weaknesses
1. How is the generalization of multi-agent policies defined? How does the generalization in this paper differ from the definitions of generalization in related works [1][2]? I recommend that the paper clearly highlight these differences in the Related Works section.
2. The motivation of the paper is somewhat unclear and requires further elaboration. In particular, the statement in Section 2 (Related Works) that "However, these approaches are not specifically tailored for Contextual MARL tasks." needs more clarification. From my understanding, the aim of the paper is to address the generalization problem by proposing a context-based learning approach for MARL. However, the rationale behind choosing this approach, as well as a detailed discussion on the shortcomings of existing MARL methods in handling generalization, are lacking and need to be presented in a more systematic manner.
3. What does the 'draw' operation in Equation (5) mean? Additional clarification should be provided here.
4. The experimental setting of the paper needs to add relevant experiments. ①The proposed method seems to adopt a role-based assignment approach, and it would be useful to compare it with works like ROMA and RODE. ②Furthermore, the Action Regularization method discussed in the paper does not impose any constraints on the action space. I am curious about how this ablation method performs on the SMACV2 benchmark.
5. Why does Zerg perform worse than Terran and Protoss in Figure 4? I don't understand why this phenomenon occurs, as there shouldn't be a significant difference at the algorithmic level. Simply stating in the paper that 'the results derived from integrating CAID into QMIX were constrained by QMIX's suboptimal credit assignment capabilities' is not sufficient. It would be more convincing to include additional experiments here.

References

[1] Mahajan, Anuj, et al. "Generalization in cooperative multi-agent systems." arXiv preprint arXiv:2202.00104 (2022).

[2] Qiu, Wei, et al. "Rpm: Generalizable multi-agent policies for multi-agent reinforcement learning." In ICLR. 2023.

**Questions For Authors:**

See the section “Other Strengths And Weaknesses”.

**Relation To Broader Scientific Literature:**

No.

**Theoretical Claims:**

I have checked the correctness of any proofs for theoretical claims.

---

> ### Author Rebuttal · Authors · 2025-03-31
>
> We sincerely thank the reviewer for the thoughtful and detailed comments. Below we respond to the key points raised in the "Other Strengths and Weaknesses" section:
>
> **Q1**: How is policy generalization defined? How does it differ from definitions in [1] [2]?
>
> **A1**: Thank you for this important question. In Section 3.3, we formalize a new generalization framework called *Contextual Multi-Agent Reinforcement Learning (CMARL)*. Unlike traditional multi-task or multi-agent generalization settings (as in [1] [2]), CMARL assumes that different environmental states across episodes can be aligned through latent contextual variables (e.g., rotation, mirroring, permutation). Our goal is to unify semantically equivalent contexts into a compact representation and dynamically assign identities to corresponding agents. This enables policy reuse under *context shift* rather than *task shift*. We will revise the Related Work section to clearly distinguish CMARL from the generalization definitions in [1] [2].
>
> **Q2**: The motivation is unclear and the limitations of prior work are not systematically discussed.
>
> **A2**: We appreciate your constructive feedback. We will clarify our motivation by systematically identifying limitations in existing MARL methods: (1) They typically fail to capture semantic transformations in contextual environments (e.g., agent position mirroring), and (2) they lack dynamic identity alignment, making it difficult to reuse policies across variants. We will make explicit that CAID addresses these issues by generating contextual representations and dynamically consistent agent identities for improved generalization.
>
> **Q3**: The meaning of the “draw” operation in Equation (5) is unclear.
>
> **A3**: Thank you for pointing this out. “Draw” denotes sampling from the agent identity distribution computed via the decoder. We apply Straight-Through Gradients to enable gradient backpropagation. We will revise the equation and add detailed explanations in the main text.
>
> **Q4**: Missing comparison with ROMA/RODE; no ablation on Action Regulator.
>
> **A4**: Thank you for these insightful comments. We address them as follows:
>
> ① We have added comparisons with ROMA and RODE (See the figure in https://anonymous.4open.science/r/CAID-7A6C/Role_methods.jpg). CAID still outperforms these methods on SMACv2.
>
> ② We carried out the ablation variant “CAID w/o AI” (without Action Regulator) on SMACv2. Results (See the left figure in https://anonymous.4open.science/r/CAID-7A6C/Ablation_study.jpg) show that removing this module significantly degrades performance, confirming its necessity for consistent action semantics under identity shift.
>
> **Q5**: Zerg performs worse than other races in Figure 4.
>
> **A5**: Regarding the Zerg performance drop, we believe that Zerg units have higher variance in attack patterns and durability, which makes the value decomposition more sensitive to alignment. As the foundational algorithm of CAID, QMIX exhibits greater performance degradation under these conditions due to its suboptimal credit assignment capability.
>
> Thank you again for your valuable suggestions. They have greatly helped us improve the clarity and rigor of our work. And we also appreciate it if you have any further comments.

---

### Official Review · Reviewer_v3VM · 2025-03-13

**Overall Recommendation:** 4

**Summary:**

The authors introduce Context-Aware Identity Generation framework which is able to generalize between tasks in one Contextual MARL domain. CAID integrates dynamically assigned identity information into action decoding for each agent, which is claimed to provide smooth adaptation to varying contexts. Combined with Action Regulator, which uses identities to produce actions from agents, and Contextual State Encoder, which encode MARL interaction history to context sequence, it was shown that CAID outperforms various baselines on classic MARL environments, such as StarCraft SMAC, Vectorized Multi-Agent Simulator and Traffic Signal Control environments. Moreover, authors compare CAID without components and show that each component is important for better performance.

**Claims And Evidence:**

The main claims are supported with experiments: it is clear that CAID outperforms various methods, and that all three components of CAID are important for CAID performance. But it wasn’t shown that CAID roles dynamic assignment is more optimal than role assignment from methods introduced in Related Work (ROMA, RODE, COPA). Moreover, authors compare CAID with only RIIT, COLA and VMIX methods, which don’t use role assignments for more effectiveness: COLA uses consensus builder, which utilizes DINO to learn labels, which does not imply any identification learning.

**Essential References Not Discussed:**

The paper covers similar literature well, however, I did not quite understand the part about the meta-RL reference. In my opinion, paper in general discusses agent’s dynamical identity management, and it is the main feature of CAID algorithm, so I think the part about meta-RL is redundant in context of CAID algorithm.

**Experimental Designs Or Analyses:**

As noted in Methods and Evaluation criteria paragraph, all experiments designs sounds valid. Authors provide detailed description of hyperparameters analysis in appendix A. On the other hand, there is some misunderstandings on analysis of experiments, because ablation study shows only importance of identity decoder compared to CAID without identity decoder, so it does not show in full effectiveness of dynamical identification learning in comparison with previous methods.

**Methods And Evaluation Criteria:**

Authors compare CAID with lots of baselines on a large number of datasets, making comparison fair with calculating experiments with 5 seeds, and setting up hyperparameters according to original methods. Evaluations are fair and make sense for the problem.

But I also wrote my concern about evaluation methods in Claims and Evidence, it seems better to compare CAID with ROMA, RODE or COPA, to make effectiveness of dynamic role assessment more clear

**Other Comments Or Suggestions:**

None.

**Other Strengths And Weaknesses:**

**Strengths:**

- This paper is well-written, and easy to follow
- This paper provides good experiment setup and different baselines co compare CAID with
- This paper provides intriguing CAID architecture with interesting insights of identity use in MARL agents
- This paper provides detailed experiments about importance of each part of CAID

**Weaknesses:**

- This paper does not provide experiments with baselines uses different strategies of identity management (ROMA, RODE, COPA), and does not show effectiveness of proposed identity management in comparison to previously proposed.

**Questions For Authors:**

In Related Work, authors also have a paragraph of discussing Meta Reinforcement Learning. I suggest that authors claim that CAID can adapt to similar tasks in one MARL domain, but authors did not highlight it clearly in abstract and experiments. Do the authors think that CAID is able to adapt to tasks in one MARL domain, and maybe authors can confirm it with experiments? I thought it was wrong to deduct points for this claim, as it was not highlighted clearly in abstract, but if authors also propose it, it would be better to provide experiments for multitask adaptation of CAID.

**Relation To Broader Scientific Literature:**

The paper is related to MARL literature and propose novel and intriguing architecture for managing MARL agents roles compared to previous architectures.

**Theoretical Claims:**

Paper does not provide complex theoretical claims, and those that are provided are correct. Authors provide detailed description of their method, so I don’t have questions about CAID methodology.

---

> ### Author Rebuttal · Authors · 2025-03-31
>
> We sincerely thank the reviewer for the detailed and constructive comments! We are grateful for your careful reading of both the main paper and the supplementary material. Below, we respond to your concerns point by point:
>
> **Q1**: The paper does not compare with dynamic role assignment methods.
>
> **A1**: Thank you for raising this important point. In the initial submission, we focused on comparisons with RIIT, COLA, and VMIX, as these methods are widely adopted and perform strongly on the newly CMARL benchmarks such as SMACv2. However, we fully acknowledge the significance of role-based methods like ROMA and RODE in learning strategies. To address your concern, we have included experimental comparisons with ROMA and RODE (See the figure in https://anonymous.4open.science/r/CAID-7A6C/Role_methods.jpg). Our preliminary results indicate that CAID achieves superior performance and faster convergence, especially in environments with contextual variations such as agent permutations, type switches, and initial state shifts.
>
> **Q2**: The relevance of Meta-RL references is unclear.
>
> **A2**: We appreciate your thoughtful observation. We originally referenced Meta-RL to provide context for generalization strategies in reinforcement learning. However, we agree that CAID does not follow the standard meta-learning paradigm. Rather, it addresses the generalization challenge by generating context-aware agent identities that unify behaviors across task variants. To improve clarity and focus, we will revise the Related Work section to remove or rephrase the Meta-RL discussion, better highlighting CAID's distinct contribution in dynamic identity generation.
>
> **Q3**: Does CAID support multi-task adaptation within the same MARL domain?
>
> **A3**: Thank you for this question. As introduced in Section 3.3 of our paper, CAID is specifically designed for the *Contextual Multi-Agent Reinforcement Learning (CMARL)* setting. While CMARL may involve variations across episodes—such as changes in agent initial positions, or types—it differs fundamentally from traditional multi-task reinforcement learning. In CMARL, the core assumption is that tasks share a common structure and can be semantically aligned through a latent contextual representation. The objective is not to learn separate policies for separate tasks, but rather to enable policy reuse by reasoning over contextual variations. Thus, CAID is built to tackle *context shift* rather than *task switch*. We appreciate the reviewer for raising this important point and will clarify the distinction between CMARL and conventional multi-task RL more explicitly in the revised paper.
>
>
> We really appreciate your comments and they really help us improve our paper! And we also appreciate it if you have any further comments.

---

> > ### Comment · Reviewer_v3VM · 2025-04-04
> >
> > Dear authors!
> >
> > I want to thank you for a quality of your answers! I was satisfied by your comparison for all the requested methods RODE and ROMA. After reading the other reviews and rebuttals, I can see that you provide lots of additional experiments showing effectiveness of CAID role assignment. Especially experiment with different role assignment strategies is strongly important in context of provided CAID method.
> >
> > It's good that you clarify CAID is not follow the standard meta-learning paradigm. I think that it would be nice if you higlight this difference clearly in further iterations of the paper. However, overall contribution of CAID method is great. I would like to increase my score to Accept.

---

> > > ### Author Response · Authors · 2025-04-05
> > >
> > > Thank you very much for your thoughtful and encouraging feedback! We completely agree with your suggestion to highlight this difference more clearly in the final version, and we will make sure to revise the introduction and related work sections accordingly.
> > >
> > > We sincerely appreciate your support and the time you invested in carefully reviewing and engaging with our work. Your comments and score adjustment mean a lot to us!

---

### Decision · Program_Chairs · 2025-05-01

**Decision:**

Accept (poster)

**Comment:**

This paper introduces Context-Aware Identity Generation (CAID), a novel framework designed to improve generalization in multi-agent reinforcement learning within the Contextual MARL (CMARL) setting, where task configurations vary across episodes. CAID employs a causal Transformer to dynamically generate agent identities based on context and integrates these identities into the value function estimation process.

Initial reviews were mixed, generally appreciating the novelty and empirical results on benchmarks like SMACv2 but raising valid concerns. These included the need for comparisons against established role-assignment methods (like ROMA/RODE), more extensive ablation studies to validate the contribution of different components (especially the dynamic identity generation), clarity on the specific CMARL problem setting versus other generalization frameworks, and requests for analysis of computational overhead and interpretability of the learned identities.

The authors provided a comprehensive and convincing rebuttal. They furnished additional experimental results comparing CAID against ROMA and RODE, performed the requested ablation studies (demonstrating the value of dynamic identities over fixed or no identities), offered t-SNE visualizations suggesting meaningful identity/context representations, and clarified the CMARL definition, the Action Regulator mechanism, and computational costs. This rebuttal successfully addressed the primary concerns of three reviewers (v3VM, 139G, Tgnb), who subsequently raised their scores or indicated their intention to do so, moving towards acceptance. One reviewer (PyYp) remained unconvinced, maintaining concerns about the precise definition of generalization used and the method's applicability, although their engagement in the discussion period after acknowledging the rebuttal was limited compared to the others.

Considering the substantial positive response to the authors' thorough rebuttal, the additional evidence provided, and the resulting majority support from the reviewers who found their initial concerns resolved, the paper makes a solid contribution to addressing generalization in MARL.